



# Cloud responses to climate variability over the extratropical oceans as observed by MISR and MODIS

Andrew Geiss, Roger Marchand

Department of Atmospheric Sciences, University of Washington,
Seattle, Washington, 320 Atmospheric Sciences—Geophysics Building
Box 351640, Seattle, Washington, 98195-1640

*Correspondence to:* Andrew Geiss (avgeiss@gmail.com)

**Abstract.** Linear temporal trends in cloud fraction over the extratropical oceans, observed by NASA's Multiangle
Imaging Spectro-Radiometer (MISR) during the period 2000-2013, are examined in the context of coincident
ECMWF reanalysis data using a maximum covariance analysis. Changes in specific cloud types defined with
respect to cloud top height and cloud optical depth are related to trends in reanalysis variables. A pattern of reduced
high altitude optically thick cloud and increased low altitude cloud of moderate optical depth is found to be
associated with increased temperatures, geopotential heights, and anticyclonicity over the extratropical oceans.
These and other trends in cloud occurrence are shown to be correlated with changes in the El Niño Southern
Oscillation, the Pacific Decadal Oscillation, the North Pacific Index, and the Southern Annular Mode.

## 1 Introduction

Clouds play a fundamental role in Earth's climate due to their effect on the planet's radiative budget. Cloud
responses to climate change are poorly understood however, and cloud-climate interaction is presently one of the
largest sources of uncertainty in climate models (Caldwell et al., 2016; IPCC, 2013; Bony et al., 2006). Several
changes in midlatitude cloud are expected under global warming with varying degrees of certainty: including
poleward shifts in the storm tracks, rising melting level, rising high cloud tops, and reduced low cloud (IPCC, 2013).
Understanding changes in midlatitude and Southern Ocean cloud in-particular is important, because these clouds
have a large radiative impact, influence atmospheric dynamics (Kay et al., 2016; Hwang and Frierson, 2013), and
are not adequately captured by climate models (Trenberth and Fasullo, 2010; Bodas-Salcedo et al., 2014).   Several
studies have observed midlatitude cloud responses to extratropical synoptic variability, for instance, changes in
cloud cover associated with the North Atlantic Oscillation or Southern Annular Mode (e.g. Li et al., 2014(a);
2014(b); 2016; Ceppi and Hartmann, 2015; Gordon and Norris, 2010; Gordon et al., 2005; Tselioudis et al., 2000).
Many studies of cloud variability (including several of those cited above) are based on analysis of International
Satellite Cloud Climatology Project (ISCCP) datasets. ISCCP is a multi-instrument and multi-satellite product that
combines observations from polar orbiting and geostationary weather satellites to determine cloud amount and
categorizes clouds by their cloud top pressure and optical depth (Rossow et al., 1999). For example, Bender et al.
(2011) use meridional maxima in ISCCP total cloud fraction as a proxy for midlatitude storm track latitude, and
identify a 25-year poleward trend in storm track location. Building on early studies, Norris et al. (2016) examine
trends in the ISCCP and Extended Pathfinder Atmospheres datasets after applying several empirical corrections
(Norris and Evan, 2015). They also show a poleward trend in storm track location between the 1980's and 2000's



and identify an increase in global cloud top height. These findings are consistent with Hadley cell expansion and increased tropopause height predicted by climate models under increased $CO_2$ (IPCC 2013).

While these studies are insightful, the ISCCP data suffer from difficulties related to changes in instrumentation, orbital configurations, viewing angle, and individual instrument calibration drifts that make it difficult to use these data to identify subtle changes in cloud fraction, especially for particular cloud types. Authors of these previous studies recognized and used various approaches to account for limitations of the data (Evan et al., 2007; Bender et al., 2011; Norris and Evan 2015).  Only recently have higher quality cloud datasets from NASA's Earth Observing System (EOS) reached sufficient duration to begin to observe trends on intra-decadal time-scales. Marchand (2013) found that both the Multi-Angle Imaging Spectro-Radiometer (MISR) and the MODerate resolution Imaging Spectro-radiometer (MODIS) datasets contain linear temporal trends in cloud fraction (for specific cloud types) in various regions between 2000 and 2010. These trends include: (A) a decrease in mid-level clouds over the Southern Oceans (30º to 50º S and strongest over the South Pacific), (B) a decrease in high-level clouds and increase in low level clouds over the North Pacific, and (C) an increase in cloud of moderate optical depth (1.3<OD<23) and an approximately compensatory reduction in cloud of high optical depth (OD>23) over the extratropical oceans of both hemispheres.

To better understand the MISR and MODIS extratropical cloud trends, we explore linkages between the MISR extratropical cloud fraction trends and trends in ERA-Interim reanalysis variables. We do this using a Maximum Covariance Analysis (MCA) to identify spatially coincident trends in cloud occurrence and several reanalysis variables. Furthermore, we examine correlations with known modes of climatic variability including the El Niño Southern Oscillation, the Pacific Decadal Oscillation, and the Southern Annular Mode.

## 2 Data

### 2.1 Cloud Observations

We use the Multiangle Imaging Spectro Radiometer (MISR) L3 1-degree gridded CTH-OD Version 6 dataset, from 2000 to 2013 (Marchand et al., 2010). MISR is a polar orbiting instrument aboard EOS Terra. MISR images the Earth using an array of nine push-broom style cameras with four spectral bands (red, green, blue and near IR) oriented at different viewing angles along its orbit, and uses multi-view geometry to estimate cloud top heights (and several other products) (Diner et al., 1998). In this dataset, total cloud cover (or total cloud fraction) in each spatial grid cell is divided into a Cloud Top Height versus cloud Optical Depth (CTH-OD) joint histogram. Cloud longwave radiative effects are directly related to cloud top temperature which is related to cloud top height, while cloud shortwave radiative effects are directly related to cloud optical depth. As such, the CTH-OD joint histograms are convenient for studying cloud-climate interaction because they categorize clouds based on physical properties which relate to their expected radiative effects. Here, "cloud fraction" is defined as the fractional occurrence of cloud (for any histogram bin or combination of bins) at a given location in a month relative to the total number of observations (pixels) observed by MISR. MISR observes most of the Earth every 8-days, and has a resolution (at nadir) of about 275 m from which cloud occurrence (detection), cloud-top-height and cloud-optical depth are determined over ice-





free ocean on a 1.1 km grid.  A detailed description of the MISR CTH and OD retrievals can be found in Marchand et al. (2010).  Each L3 cell in the monthly 1-degree grid typically has several thousand observations associated with it, though many of these will have been taken concurrently (with a high degree of correlation). At 1-degree resolution, in the extratropics, the MISR cloud occurrence data has a zonal autocorrelation length of less than 5-degrees, and we perform our analysis using 5-degree spatial averages of the MISR data (such that each of our grid cells is effectively an independent sample).

Trends computed from MODerate resolution Imaging Spectro-radiometer (MODIS) collection 6, 1-degree gridded Cloud Top Pressure and Optical Depth histograms (Hubanks et al., 2015) are briefly compared with MISR trends. MODIS is a 36-band polar orbiting radiometer aboard both EOS Aqua and Terra, with equatorial crossing times of about 10:30 and 13:30 respectively. MODIS provides cloud fraction joint histograms that are similar in structure to the MISR joint-histograms, but are retrieved using different algorithms, which are described by Platnick et al. (2015) and compared with MISR and ISCCP retrievals in Marchand et al. (2010).  In MODIS collection 6 processing (Platnick et al., 2017), pixels that are determined to be only partly cloudy or on the edge of a cloud (meaning cloudy pixels that border a clear pixel) are stored in a separate histogram from other cloudy pixels, which are nominally fully cloudy.  Inclusion of the partly cloudy and edge pixels does not have a substantial effect on the MODIS cloud fraction trends. Because the quality of partly cloudy and edge pixel retrievals is suspect (and for consistency with Marchand 2013 who used the MODIS collection 5 cloud product that does not include partly cloudy or edge pixels), we show results without the partly cloudy or edge pixels in later figures.

**2.2 Reanalysis**

The ERA-Interim reanalysis (Dee et al., 2011) is maintained by the European Center for Medium Range Weather Forecasts (ECMWF). The data used here span the same period as the MISR data (March 2000 – March 2013) and are the ERA-Interim "monthly means of daily means." The MISR cloud fraction data are compared to several reanalysis variables: temperature ($T$), geopotential ($Z$), specific humidity ($q$), vertical velocity ($\omega$), divergence ($\nabla$), absolute vorticity ($\zeta$), Sea Surface Temperature ($SST$), and Sea Level Pressure ($SLP$). The data used are defined at 20 pressure levels: 1000-50 hPa by increments of 50. Because MISR retrieves physical cloud top height, as opposed to cloud top pressure, and because the ECMWF data are defined at pressure levels, the hypsometric equation was used to derive altitudes ($z$) for the monthly ECMWF data, and the reanalysis data were then linearly interpolated onto MISR's cloud top height grid.

It should be noted here that potentially spurious features exist in the reanalysis specific humidity trends. The ERA-Interim data indicate a ubiquitous increase in low level tropospheric specific humidity nearly everywhere, but particularly in the tropics, which is on the order of 0.1-0.3 g kg$^{-1}$ decade$^{-1}$, and occurs primarily below 750hPa. We performed a brief comparison to specific humidity data from the Modern Era Retrospective analysis for Research and Applications (MERRA) and found that while it did indicate an increase in low level specific humidity of similar magnitude in some of the Northern Hemisphere, it did not corroborate the pervasive trend in the ECMWF data. Dessler and Davis (2010) provide a more comprehensive inter-comparison of specific humidity trends in different reanalysis datasets, albeit not for the time-period analyzed here, and conclude that while most show recent increases



in specific humidity (which is expected under global warming), there is large disagreement over the magnitude and spatial distribution. In any case, we show the ECMWF specific humidity data, but caution the reader that these data may not be as robust or reliable as other fields.

**2.3 Climate Indices**

Several of the patterns found via the MCA (Sect. 4-6) resemble well-documented modes of climate variability. In Section 4, we compare the monthly time series of the MCA modes to monthly indices for various modes of climate variability and northern hemisphere teleconnection patterns maintained by the NOAA Climate Prediction Center (CPC). The indices used are:

- The Niño Region 1+2, 3, 3.4, and 4 indices (hereafter referred to as "Niño 3.4," for instance). These indices are based on spatial averages of SST anomaly in various regions in the tropical Pacific Ocean:

  1+2: Latitude: 0°-10°S, Longitude: 90°-80°W

  3: Latitude: 5°N-5°S, Longitude: 150°-90°W

3.4: Latitude: 5°N-5°S, Longitude: 170°-120°W

  4: Latitude: 5°N-5°S, Longitude: 160°E-150°W
- The Pacific Decadal Oscillation (PDO) index, which is the first mode of an Empirical Orthogonal Function (EOF) decomposition of SST north of 20°N in the Pacific Ocean (Mantua et al., 1997).
- The Antarctic Oscillation index or "Southern Annular Mode" (SAM), which is defined as the first
mode of an EOF analysis of 700hPa geopotential height south of 20°S 1979-2000 (Thompson and Wallace, 2000).
- The Pacific-North American (PNA) mode index. The index is defined by projection of the PNA loading pattern on to the daily 500hPa height anomalies over the entire Northern Hemisphere. The PNA loading pattern is derived by a rotated principal component analysis of 500hPa heights north
of 0° between 1950 and 2000 as described in (Barnston and Livezey, 1987).
- The North Pacific Index (NPI). The NPI is a standardized mean of sea level pressure between 30°-65°N and 160°-220°E (Trenberth & Hurrell, 1994), and was obtained from the University Corporation for Atmospheric Research (UCAR) website (link provided at bottom of the reference list).

Several other similarly defined teleconnection patterns were analyzed, but are less relevant to our results: the East Pacific/North Pacific, Scandinavian, Tropical / Northern Hemisphere, East Atlantic, Pacific Transition, Polar / Eurasia, and West Pacific indices. All indices but the NPI were obtained from the NOAA CPC website (see data availability section). These indices are dimensionless except for the "Niño" indices, which represent temperature anomalies, though we have standardized each index prior to computing any statistics by subtracting the mean,
dividing by the standard deviation, and detrending.



### 3 Trends in Cloud Fraction

Linear temporal trends were computed on the MISR cloud fraction data. Figure 1 shows the trends computed in each of four extratropical ocean basins for each bin in MISR's CTH-OD joint histograms in panels (a)-(d). The four

regions studied are the North Atlantic (25-65N, 280-360E), North Pacific (25-65N, 120-240E), South Atlantic (25-65S, 280-360E), and South Pacific (25-65S, 120-240E), (these geographic regions are shown in Figs. 2 and 3, which show trends in ERA interim data and results of the MCA that is discussed in Section 4). The cloud fraction data were spatially averaged over each ocean basin prior to computing trends, and the composited seasonal cycle was removed. The middle row of panels (e)-(h) in Fig. 1 show the MISR cloud fraction trends associated with each

optical depth category, here cloud fraction is summed with respect to cloud top height prior to computing trends. Finally, the bottom row of panels (i)-(l) show cloud fraction trends for each MODIS optical depth category for both MODIS Aqua and Terra. Note that the MODIS cloud occurrence histograms do not include a "No Retrieval" (NR) category. The MODIS data also include separate bins for optical depths between 60-100 and 100-150 (the high optical depth bin is new for MODIS collection 6), however in Fig. 1, these two optical-depth bins have been

summed to create a single bin representing all optical depths greater than 60. This step is taken to make comparison of the two datasets easier. Bold bordered bins in the joint histograms in the top row (a)-(d) indicate that the cloud fraction trend in that bin exceeds a 95% confidence test, while the bars in the lower panels indicate the 95% confidence interval. The confidence intervals were computed using a windowed boot-strapping technique described in Wilks (2006), which was also used to assign confidence to the cloud fraction trends computed in Marchand 2013.

This technique involves randomly resampling, with replacement, each bin's cloud-fraction time-series in 12-month chunks 1000 times and computing trends for each of the resampled time-series. The trend associated with the original time series is said to be significant at the 95% level if it exceeds the 25th most positive or 25th most negative resampled trend. Bins that account for less than 0.1% of the total cloud fraction are not considered. Figure 1 shows the same dominant pattern of changing extratropical cloud fraction identified in Marchand (2013), but here based on

three additional years of MISR data. The results in three of the basins (the North Pacific, South Atlantic, and South Pacific) are characterized by increasing cloud of moderate optical depth, particularly at low altitudes, and decreasing cloud of high optical depth at most levels (with the largest decreasing trends at high levels). The South Atlantic differs in having no clear increase in low-level clouds, only a reduction in optically thick clouds. The South Pacific also differs from the other basins, featuring a stronger reduction in mid-level cloud (between 2 and 5 km).

There are no significant changes in the occurrence of MISR failed optical depth retrievals (that is the NR column), but there is a slight reduction in the number of failed cloud-top-height retrievals (that is the NR row) in the North Pacific and North Atlantic. Failed cloud-top-height retrievals most often occur in multilayer cloud conditions (where a low cloud that is visible in the MISR nadir view cannot be located in an off-nadir views due to a visibly opaque or semitransparent higher altitude cloud), which suggest that a small portion of the observed increase in low-

level clouds is due to a reduction in higher (semi-transparent) cloud.

We note here that an analysis of the MISR calibration suggests that MISR near-IR radiances (used to obtain MISR optical depths) likely have a small downward drift of about 0.9 to 1.5 %10yr⁻¹ (Bruegge et al., 2014; Corbett and



Loeb, 2015; Limbacher and Kahn, 2016). This calibration drift can be expected to reduce the retrieved optical depth, reducing the occurrence of clouds with large-optical depths in the CTH-OD product, and increasing the

occurrences of clouds with moderate optical depths. Such a calibration drift will not change the cloud top altitude but will cause clouds at a given altitude to shift toward lower optical depths at that same altitude. Evidence for such a calibration-driven change can be seen to some degree in Fig. 1, where in the South Pacific (panel (d)) and South Atlantic (panel (c)) between 5 and 7 km there is a strong increase of cloud with OD between 9.4 and 23, and strong decrease at this same altitude for optical depths greater than 60. However, Fig. 1 suggests that much of the

reduction in optically thick cloud is occurring at high altitudes while the increase in clouds with moderate optical depths is occurring for low level clouds. Limited testing, where the CTH-OD dataset was reprocessed for one-month with the observed radiances reduced artificially by 2%, suggests a reduction in the occurrence of cloud with large optical depths may well explain 50 to 75% of the MISR trend depending on the region. Plans are underway to reprocess the entire MISR mission, starting from the (level 1) calibrated imagery and eventually including all

higher-level datasets (level 2 swath and level 3 global datasets), and this includes the CTH-OD product. This reprocessing will include a correction for this calibration drift among other issues. For the present, an important caveat is that the strength of the optical depth trend in the MISR dataset (and associated statistical confidence) is likely being overestimated, and this may explain why the trends in the MISR dataset are larger than MODIS. The significant changes in cloud top height observed in three of the study regions and the reduced mid-level cloud

(between about 2 or 2.5 and 4 km) observed in the South Pacific could not have been caused by this calibration drift however.

In the bottom panels (i)-(l) of Fig. 1, trends in MODIS cloud fraction bins only partially corroborate those in the MISR dataset. MODIS Aqua identifies a reduction in optically thick cloud in all the regions studied but the South Pacific, while MODIS Terra shows a reduction in only the North and South Atlantic. MODIS Aqua shows an

increase in cloud of moderate optical depth in all regions but the South Atlantic (in agreement with MISR) while MODIS Terra shows little or no change in these bins. As with MISR, there is evidence for drifts in the MODIS calibration. Corbett et al. (2015) compares both MISR and MODIS Terra level 1 radiances to collocated CERES outgoing shortwave radiation observations and finds that while MISR red green and near IR bands have darkened relative to CERES, MODIS Terra red and near IR bands have brightened, which likely explains much of the

discrepancy between MODIS Terra and MISR cloud optical thickness trends. Taken in combination, the MISR and MODIS data suggest there has been a reduction in cloud optical thickness during the period studied, at in least in the North and South Atlantic Ocean and likely the North Pacific as well.

As another note, we add that there is a known issue with the MODIS Terra cloud mask over ocean. MODIS Terra's 8.6 μm channel has undergone warming since around 2010 that has not been corrected though on-board calibration,

and this warming has caused a number of clear pixels to be flagged as cloudy in the MODIS cloud product. This problem primarily affects the low cloud retrieval fraction in tropical and subtropical regions with low average total cloud fraction and does not appear to have a substantial impact on the MODIS Terra extratropical cloud trends shown in Fig 1. This trend is being corrected in MODIS collection 6.1.



On monthly time-scales cloud occurrence is heavily influenced by synoptic conditions, and it seems likely that a
large portion of the observed cloud fraction trends are related to trends in synoptic variables. In Fig. 2, we show the
spatial distribution of trends in several of the reanalysis variables discussed in Section 2 (500hPa geopotential
height, temperature, and absolute vorticity), hashing denotes trends that are significant with 95% confidence (using
5x5-degree bins). These trends have been computed after first de-seasonalizing the data using compositing, and
confidence intervals are determined using the windowed bootstrapping technique discussed above. There is a
notable increase in both the 500hPa temperature and geopotential height in the center of the North Pacific region
(Fig. 2 (d)-(e)). This is accompanied by increased anticyclonicity (f) (a negative trend in absolute vorticity in the
northern hemisphere), which is expected given the temperature and pressure changes, though the trends in
anticyclonicity often do not pass the significance test, which is perhaps due to the large variability in this field. Both
the 500hPa temperature and absolute vorticity fields show increasing trends in both Southern Ocean storm track
regions (around 40-50S), with only small areas that pass the confidence test which coincide with the largest positive
trends (panels (h), (i), (k), (l)). These are accompanied by an increase in geopotential heights that does not pass the
95% significance test, but is meteorologically consistent with the corresponding changes in temperature and
vorticity (panels (g) and (j)). In the South Pacific this primarily occurs to the south and the east of New Zealand. In
the South Atlantic there is a meridional dipole with the strongest positive changes in these fields in the poleward part
of the region. Notably, the regions in the Southern Ocean that show increased geopotential heights, temperature, and
anticyclonicity correspond well with the loading pattern of the Southern Annular Mode which has undergone a
positive trend during the study period and will be discussed in more detail later (Sections 5.3 and 5.4). In the North
Atlantic, while there is a weak increase in anticyclonicity (c), as well 500 hPA geopotential height (a) and
temperature (b), along the storm track (which is similar in sign to the North Pacific) the trends are generally smaller
than in the other three basins, with essentially no significant trends.

In summary, in the ocean basins studied, ERA-Interim reanalysis shows increased temperature, geopotential heights,
and anticyclonicity in the storm track latitudes, though more poleward in the South Atlantic. The largest and most
robust changes are in the central North Pacific Ocean and weakest in the North Atlantic. Except perhaps in the North
Pacific, most of the trend (linear fit to the change) in the reanalysis variables is not statistically significant, however
we stress that this doesn't mean that there has been no change in meteorology or that these changes have no impact
on the clouds.  To the degree the reanalysis data is accurate there has been an increase in temperature, geopotential
heights, and anticyclonicity in some portion of each of these basins over the period examined. Rather, the lack of
statistical significance means that relative to the annual variability, the change is small and could be a result of
annual variability rather than a true trend in the mean temperature, geopotential heights, and anticyclonicity with
time. In the following section we use a maximum covariance analysis to identify linkages between trends in the
reanalysis variables and the MISR cloud fraction data. Finally, we show that the changes discussed above are
consistent with recent trends in the North Pacific index and the Southern Annular mode and explore correlations
with other climate indices.





**4 Maximum Covariance Analysis**

We wish to identify relationships between two high dimensional datasets. The MISR dataset is a function of time, latitude, longitude, cloud top height and cloud optical depth, while the ERA interim variables described in Section 2 vary in time, height, latitude, and longitude. In particular, we wish to identify which trends in cloud occurrence observed by MISR are related to trends in meteorology. To do this, we identify patterns of trending cloud

occurrence, in the context of MISR cloud occurrence histograms, that tend to be co-located with patterns of trending meteorological variables using Maximum Covariance Analysis (MCA), the result of which are shown in Fig. 3. MCA falls into the family of other linear decomposition techniques such as Singular Value Decomposition (SVD), principal component analysis, empirical orthogonal function analysis, canonical correlation analysis and others, (Bretherton et al., 1992; Hannachi et al., 2007; Von Storch and Zwiers, 2001). Typically, in climate science, SVD

(often interchangeably referred to as empirical orthogonal function analysis and principal component analysis) is applied to a single time-varying field to identify important spatial patterns that co-vary in time. This involves decomposition of the field's time-covariance matrix. SVD was used in the atmospheric sciences as early as Lorenz (1956) and Kutzbach (1967), though it became significantly more popular in the 1980-90's when it was used to identify a number of well-known modes of climate variability, such as the North Atlantic Oscillation (Barnston and

Livezey, 1987; and Wallace and Gutzler, 1981), the Pacific Decadal Oscillation (Mantua et al., 1997), and the Antarctic Oscillation (Thompson and Wallace, 2000). MCA involves applying SVD to a cross-covariance matrix computed between two datasets. It is similar to canonical correlation analysis which involves decomposition of a cross-*correlation* matrix. For instance, Prohaska (1976) applies SVD to the temporal correlation matrix computed between monthly mean sea level pressure and temperature fields. MCA has also been used historically to understand

climate data, in particular, since about 2000, it has been heavily used to identify interactions between sea surface temperature and atmospheric variables (Czaja and Frankignoul, 1999; Liu et al., 2006; Frankignoul et al., 2011, and references therein). It has been used in other areas of the earth sciences as well, for instance in sea ice modeling (Dirkson et al., 2015) and study of the structure of the Madden Julian Oscillation (Adames and Wallace, 2014).

A complete mathematical description of the formulation used here, as well as a discussion of the significance of

MCA modes, is given in Appendix I, and results are shown in Fig. 3.  In short, a covariance matrix is computed that represents spatial covariance between trends in the MISR and trends in the ERA variables. A Singular Value Decomposition (SVD) is applied to the covariance matrix. One set of singular vectors identified by the SVD then represent patterns (or modes) of cloud trends in CTH-OD space (shown in the middle column of Fig. 3), and the other set represent corresponding patterns in trends of the vertical profiles of the reanalysis variables (shown in the

right column of Figure 3). These patterns (or modes as we will call them in later sections) can be projected on to the original trend data to see corresponding spatial patterns (the left column of Fig. 3).   In the left column of Fig. 3, bright red colors represent regions where the cloud occurrence and reanalysis trends shown in the panels to the right have occurred. The same is true for the blue colors except that in these regions the trends are of the opposite sign. For example, in North Atlantic mode 1 (the first row in Fig. 3) the red region around 45 N in the left-most panel

indicates that in the North Atlantic there has been an increase in low cloud of moderate optical depth and a reduction



in high thick cloud, and at the same location there has been increased geopotential heights, temperature, and anticyclonicity at most levels. In Section 5, we discuss the modes and their geographic patterns in detail.

Unlike an EOF analysis, the original monthly datasets cannot be fully recovered from the MCA modes alone (information is lost when trends are computed and because the SVD is performed on the spatial covariance matrix

between the two datasets). However, monthly time series associated with each mode can be derived by linearly projecting the MCA mode patterns shown in Fig. 3 on to the monthly cloud fraction anomaly data. As noted earlier, and discussed further in the next section, many, but not all, of the MCA modes resemble well documented modes of climate variability. In many respects this is not surprising since midlatitude cloud is intimately linked to the synoptic meteorology and one might therefore expect that changes in synoptic conditions captured by the CPC

indices will be correlated with changes in cloud cover. We investigate the relationships between the MCA modes and various climate indices (introduced in Section 2) using a time-correlation analysis. In this analysis, both MCA-derived time-series and the climate indices are first detrended before Pearson correlation coefficients are computed. The correlation values (which will be discussed in the next section) are tabulated in Fig. 4, along with information about any linear temporal trends in the associated CPC index time-series.

**5. Results**

We have identified several trends in MISR cloud amount over the extratropical oceans in Fig. 1. Below, cloud fraction trends in each of the ocean basins shown in Fig. 1 are discussed in terms of the MCA decomposition for that basin. We proceed through each of the four basins and examine each MCA mode using shorthand NAX, NPX, SAX, SPX to refer to MCA mode number X in the North Atlantic, North Pacific, South Atlantic, and South Pacific

respectively. Under each of the leftmost panels in Fig. 3, the percentage of covariance between the datasets and variance within each dataset that can be explained by that mode are given as: "(% covariance, % ERA-Interim variance, % MISR variance)." These values provide some additional information about the relative importance of each MCA mode.

**5.1 North Atlantic**

NA1 captures an extratropical trend of increased low-level cloud of moderate optical depth and reduced cloud of high optical depth, primarily at high altitudes. This mode is maximized along the storm track (roughly 40° to 55° N) and is associated with increased temperature, pressure, and anti-cyclonic flow, as well as divergence at the surface, convergence aloft and downward motion. Figure 2, panels (a)-(c) shows that trends in temperature, geopotential heights, and absolute vorticity in this region do not pass a 95% confidence test, however, in the following discussion

we see that similar MCA modes (modes with similar cloud, pressure, temperature, and vorticity patterns) exist in each of the other ocean basins studied where trends in the reanalysis variables are more robust. Figure 4(a) shows there is no particularly strong connection between this mode and any climate index analyzed, but rather weak correlations with the East Atlantic teleconnection patterns and with the North Atlantic Oscillation. Li et al. 2014 identifies relationships between cloud occurrence and the NAO in Cloudsat data, however the correlations shown in





Fig. 4 and the trends in the NAO and other East Atlantic teleconnection patterns are relatively weak for the time-period analyzed, so it is not clear whether changes in the NAO are responsible for this cloud fraction trend.

NA2 explains about 15% of the covariance and 12% of the variability in the ERA trends but only 4% of the variance in the MISR trends.  As such, this mode appears to be only a minor contributor to the overall (basin average) change in cloud occurrence (shown in Fig. 1).  The ERA interim profiles for NA2 show the largest trends in near surface q,
T, and divergence, mid-troposphere vertical velocity, as well as the largest SST trend of any of the North Atlantic modes. While the changes in the cloud fraction joint histogram are somewhat noisy, the strongest response is in the low-level cloud.  Interestingly, the spatial distribution to the left resembles recent trends in North Atlantic SST, with warming SSTs off the east coast of North America and cooling in the central North Atlantic (Sup. Fig. 1). It has been documented that SSTs in the Gulf Stream influence cloud fraction (Minobe et al. 2008, 2010).  Given the
apparent connection to SST (and the fact that this mode does not have a strong correlation with any CPC index), we hypothesize that NA2 is a possible link between North Atlantic SST changes and low-level cloud fraction.

NA3 is primarily a subtropical mode (notice location of red colors in left panel of Fig. 3) and is characterized by an increase in low level cloud of moderate optical depth in the southern portion of the study region. This increase in low clouds appears to be due to an increase in specific humidity and increased convergence at low levels in the ERA
Interim data set, though as noted in Section 2, we caution this increase in specific humidity may be a spurious feature in ERA interim data or may be over-estimated and is not entirely corroborated by MERRA. A somewhat similar pattern of ERA and cloud fraction trends to NA3 is seen in the North Pacific in mode NP3, but NA3 features a small reduction in mid-level cloud around 4 km and larger reduction in high-clouds above 7 km than NP3.

**5.2 North Pacific**

NP1 is an east-west dipole, showing increased low-level cloud and decreased high-level cloud of median optical depth in the East Pacific, and the opposite to the West. The spatial structure of this mode resembles the spatial structure of the PDO, and Fig. 4 indicates there is a noteworthy time-correlation between the two (r = -0.42). It has been argued that the longer time-scale PDO is a response to ENSO and other factors including variability in the Aleutian low (Schneider and Cornuelle 2005, Newman et al. 2016).  Newman et al. (2016) show that variability in
the Aleutian low as captured by the North Pacific Index (NPI) typically leads the PDO (see their Fig. 3b) by several months. Not surprisingly, we find an even stronger correlation between NP1 and the NPI (r=0.56) and to a lesser degree the PNA (r=-0.48) (which is also known to be strongly correlated to the NPI).  The panel on the right side of Fig. 3 indicates that NP1 cloud changes are associated with large changes in the thermodynamic variables near the surface, and a particularly large change in SST, with cooler near-surface temperatures and SST in the eastern third
of the North Pacific associated with increased low cloud amount, and conversely warmer SSTs being associated with less low cloud.  We note that the sign of the MCA modes is arbitrary and, as shown, is opposite that defined by the PDO (PDO is positive with warmer SST in the eastern pacific and hence the negative correlation).  NP1 also shows a weak increase in mid tropospheric down-welling and low-level divergence with convergence aloft, which may explain the reduction in high cloud, and would further enhance stability in the lower troposphere.



NP2 is a very similar mode to NA1 (note NA1 not NP1). It shows a reduction in high optically thick cloud and enhanced low cloud of moderate optical depth. The profiles of the ERA interim variables to the right in NP2 show increased temperature, pressure, and anti-cyclonic motion throughout the middle of the domain (along 45° N), and are nearly identical to the NA1 pattern. Figure 2 shows that both 500hPa temperature and pressure in this area have undergone robust positive trends. The time series derived from NP2 has high correlation with the NPI (r = .42), as

does NP1, but NP2 has much weaker correlation with the PDO than NP1. We examine the time-series associated with these two MCA modes and the PNA, PDO, and NPI in panels (a)-(b) of Fig. 5. These time-series illustrate the close connection between both of the North Pacific MCA modes and the NPI, as well as the first mode's relation to the PDO and the second mode's relation to the PNA. However, given the difference in spatial patterns between NP1 and NP2, and the poor correlation of NP2 with the PDO, we interpret the pattern in NP1 as a response to longer time

scale forcing related the PDO, while NP2 is associated with shorter time-scale synoptic variability captured by the PNA mode and the NPI. In effect, while NP1 and NP2 both feature increased low clouds and decreased high clouds (albeit with some relatively subtle difference in regards to optical depth), they have very different spatial patterns and do so for different reasons.

Finally, NP3 shows a similar subtropical pattern to NA3, with an increase in low level cloud of medium optical

depth and an increase in low level specific humidity mostly south of 40°N. This mode explains only a very small amount of the variance in the cloud fraction data (2%), and so it likely does not have a particularly meaningful relation to the cloud fraction trends, but is included because it is consistent with NA 3, and explains a large amount of covariance (12%).

### 5.3 South Atlantic

The MCA results in the South Atlantic are the least tractable of any of the regions studied. SA2 appears to be a 'correction' to SA1, with nearly identical large scale spatial patterns, (but with SA2 having more fine scale features), and nearly opposite changes in cloud fraction (except in the region between 1.5 and 2.5 km where SA1 shows little change and SA2 a decrease in cloud amount). SA1 also explains much more of the variance in the MISR cloud fraction data than SA2 (21% vs 4%). SA1 is associated with increased high pressure, temperature, and

anticyclonicity in the southeast part of the region (consistent with overall ERA-Interim trends shown in Fig. 2 panels (g)-(i). The region of enhanced high pressure in SA1 corresponds with a region of high pressure in the SAM loading pattern, and indeed SA1 shows moderate correlation with the SAM (r = .25) (Fig. 4a). SA3 explains only 7% of the covariance and 1% of the variance in the MISR data and is therefore not included in the analysis. In the NP and NA regions, we chose to include the third MCA modes because there was good agreement between the two basins and a

coherent spatial pattern. There is no such agreement in the third MCA modes in the southern hemisphere regions and the spatial patterns associated with the third modes appear noisy.

### 5.4 South Pacific

The changes captured by SP1 are related to the South Pacific Convergence Zone (SPCZ). The period studied was characterized by relatively neutral ENSO conditions, with more La-Niña like conditions later in the time series. The



position of the SPCZ is influenced by ENSO, shifting to the south and west during La-Niña conditions (Trenberth
        and Shea 1987 and Vincent 1994), and the spatial pattern of cloud fraction changes shown in SP 1 are consistent
        with the cloudy SPCZ shifting into the South Pacific study region. Figure 4 (a) further corroborates the relation to
        ENSO, showing high time-correlation between this MCA mode and the Niño region 3, 3.4 and 4 anomalies (r = -.43
        to -.50).  Figure 5 (c) show the SP1 and the Niño 4 time-series.

SP2 is most positive in the storm track region and the ECMWF profile trends shown on the right indicate enhanced
        geopotential heights, temperature, and anticyclonicity, even showing an increase in surface divergence with
        convergence aloft, similar to NA1, SA1, and NP2. These changes are associated with increased cloud of moderate
        optical depth at low levels and reduced mid-high level cloud of slightly larger optical depths. The SAM has
        undergone a positive trend in recent decades (Thompson and Wallace 2000, Thompson et al. 2011, Fig. 4 (b)),

which is at least partly responsible for (or at least is correlated with) these changes. The sign convention is such that
        positive values of the SAM index correspond to an enhanced low over Antarctica and an intensified polar vortex,
        along with increased surface pressure and geopotential heights over much of the Southern Ocean. Figure 4 (a) shows
        that the SAM has the highest time-correlation with SP 2 (r = .32), and the associated time-series are shown in Fig. 5
        (d). Arblaster and Meehl 2006 and Thompson et al. 2011 attribute most of the past trend in the SAM to

anthropogenic ozone depletion, but increased $CO_2$ concentration (under an RCP 8.5 scenario, see IPCC 2013) could
        cause this trend to continue even as Antarctic stratospheric ozone begins to recover (Thompson et al. 2011, Zheng et
        al. 2013). Hartmann and Ceppi 2014 relate changes in South Pacific reflected shortwave radiation observed by
        Clouds and Earths Radiant Energy System (CERES) Terra to a trend towards La Nina like conditions and to trends
        in zonal mean winds and the SAM (which are very strongly correlated). They note however, that it is difficult to

identify a robust relationship between the SAM and Southern Ocean cloud shortwave radiative effect trends due to
        the dominant influence of sea ice changes and of ENSO on such a short time period. Ceppi and Hartmann 2015 note
        that while cloud amount, particularly mid and high-level cloud, responds to changes in the annular modes,
        associated dynamically forced changes in cloud shortwave radiative effect may be of secondary importance to
        thermodynamically forced changes in the cloud phase.  Regardless, the cloud amount shown in SP2 and SA1 are

likely driven by changes in the SAM, and continued data collection should help isolate cloud change responses to
        the SAM.

## 7. Conclusions

        In closing, cloud datasets from EOS are now of sufficient quality and length to begin studying the response of cloud
        to synoptic variability on multi-year time scales. We have found via maximum covariance analysis a number of

linkages between trends in synoptic meteorology and trends in cloud fraction. Notably, increased low cloud of
        moderate optical depth and reduced high cloud of higher optical depth is associated with increased temperature,
        anticyclonicity, geopotential height, and subsidence in the extratropical storm track regions. We speculate that this
        could be linked to a strengthening of extratropical warm-core highs during the time-period studied, though this
        would require additional analysis of daily data to verify. The maximum covariance analysis also revealed linkages

between observed extratropical cloud fraction changes and known modes of climatic variability. Cloud changes (or



responses) associated with ENSO, PDO, NPI, PNA, and SAM were found, as well as a possible link between cloud fraction and changing Atlantic SSTs (i.e. mode NA2). As climate records from MODIS, MISR (and we hope future generations of equally, if not more, capable instruments) continue, we will gain increasing understanding of the response of clouds to synoptic variability on intra-decadal time scales. These observation-based relationships

potentially offer exciting and new ways in which we can evaluate and improve climate models, as well as further our understanding of climate change.



**8. Data Availability**

NOAA Climate Prediction Center indices are available from the NOAA CPC Website:

[www.cpc.ncep.noaa.gov]

The North Pacific Index is hosted on the UCAR climate data guide NPI website:

[climatedataguide.ucar.edu/climate-data/north-pacific-np-index-trenberth-and-hurrell-monthly-and-winter]

MISR CTH-OD product:

[http://climserv.ipsl.polytechnique.fr/cfmip-obs/Misr.html]

The MISR CTH-OD dataset does not have a DOI. Questions regarding this product can be directed to Dr. Roger

Marchand: rojmarch@uw.edu

CERES SSF 1 degree product. Loeb N, et al. (2015), NASA Langley Research Center:

[http://ceres.larc.nasa.gov/products.php?product=SSF1deg]

MODIS atmosphere global monthly 1 degree product:

[http://modis-atmos.gsfc.nasa.gov/MOD08_M3/index.html]

MODIS Aqua DOI:         10.5067/MODIS/MOD08_M3.006

MODIS Terra DOI:        10.5067/MODIS/MYD08_M3.006




**Appendix I:**

The MCA process is as follows: The MISR cloud fraction trend data were arranged into a 2-dimensional matrix ($\mathbf{M}$),

with one dimension representing latitude and longitude, and the other representing cloud optical depth and cloud top

height trend for each MISR CTH-OD component. The ECMWF trend data were also arranged into a 2-dimensional

matrix ($\mathbf{E}$). Latitude and Longitude again are represented by one dimension, while the trend variable type

(temperature, geopotential, pressure, absolute vorticity etc.) at each height ($z$) comprise the second dimension. This

is depicted in Fig. 6. Trends in each like-variable (e.g. $Z, T, w,$ CF, etc.) were pre-normalized, meaning that trends

for each variable had their mean value subtracted and were divided by their standard deviation. For instance, the

mean of all temperature trends is subtracted from each temperature trend, and then the temperature trend data is

divided by its standard deviation. A covariance matrix was then computed with respect to space, and a singular

value decomposition was performed on this covariance matrix:

$$\mathbf{U_M} \Sigma \, \mathbf{U_E^T} = \frac{\mathbf{M E^T}}{n}. \tag{1}$$

Here, $n$ is the number of latitude and longitude grid points included in the analysis. The right-hand side is a spatial

covariance matrix computed between the two sets of normalized trends. $\mathbf{U_M}$ and $\mathbf{U_E}$ contain left and right

eigenvectors produced by the singular value decomposition, where the columns of $\mathbf{U_M}$ and $\mathbf{U_E}$ represent modes of

spatial covariance in MISR CTH-OD space and in the vertical profiles of the ECMWF reanalysis variables

respectively. The matrix $\Sigma$ is diagonal, and includes the singular values associated with each mode. The spatial

patterns in each trend dataset associated with each column of $\mathbf{U_M}$ and $\mathbf{U_E}$ were retrieved by simply projecting

$\mathbf{U_M}$ and $\mathbf{U_E}$ on to their respective datasets:

$$\mathbf{V_M} = \mathbf{U_M^T M} \qquad \text{and} \qquad \mathbf{V_E} = \mathbf{U_E^T E}. \tag{2}$$

Here, the columns of $\mathbf{V_M}$ and $\mathbf{V_E}$ are the spatial patterns in their respective trend datasets associated with each of the

modes of covariance identified with the MCA. The columns of $\mathbf{V_M}$ and $\mathbf{V_E}$ were then standardized (in this case they

are divided by their standard deviation, but the mean is not removed) and again projected on to the original

dimensional versions of $\mathbf{M}$ and $\mathbf{E}$ (which we will call $\mathbf{M}^*$ and $\mathbf{E}^*$). This yields dimensionalized versions of the MCA

modes ($\mathbf{U_M^*}$ and $\mathbf{U_E^*}$) and associated normalized spatial patterns ($\mathbf{V_M}$ and $\mathbf{V_E}$) e.g.:

$$\mathbf{U_M^*} = \mathbf{M}^* \mathbf{V_M^T} \qquad \text{and} \qquad \mathbf{U_E^*} = \mathbf{E}^* \mathbf{V_E^T}. \tag{3}$$

Each of the first two or three MCA modes in each ocean basin is displayed in Fig. 3. The left-most panels show the

spatial distribution associated with each mode derived using the ECMWF trends ($\mathbf{V_E}$). These are normalized. The

spatial distributions derived using the MISR data are omitted for space, but are necessarily quite similar, though not

identical, to those shown. The middle panels show trends in cloud fraction joint histograms which are

dimensionalized ($\mathbf{U_M^*}$), and the right panels show the associated dimensionalized trends in the profiles of the various

ECMWF reanalysis variables ($\mathbf{U_E^*}$). Recall, that the true trend in the original dataset associated with each mode can

be recovered by projection of the contents of the middle or right panel ($\mathbf{U_M^*}$ or $\mathbf{U_E^*}$) on to the spatial distribution in the





left panel ($\mathbf{V_E}$). Changing the sign of all three panels would then yield the same result, and we have chosen the sign of each mode such that they most resemble the patterns shown in Fig. 1.

In ascertaining the significance of each MCA mode, it is important to examine the covariance between the two datasets and the variance within each dataset explained by the MCA mode. These values are printed in Fig. 3

underneath each spatial loading pattern in the format: "(% covariance, % ERA-Interim variance, % MISR variance)" explained. These measures give an idea of a mode's relative importance in the context of the original data. The percent covariance explained is a measure the importance of a mode relative to other modes, while the percent ERA or MISR variance explained measures the importance of a mode for explaining the trends in each particular dataset. For instance, the first and second modes in the North Pacific are quite robust because they explain a large amount of

the total covariance between the two datasets *and* explain a substantial amount of the variance in each dataset. On the other hand, the third North Pacific mode is less important, because while it explains a non-negligible fraction of the total covariance, it explains only 2% of the variance in the MISR data and thus does not project strongly on to the MISR trends. While there is no universally agreed upon method for testing the significance of MCA results, a useful metric is the "normalized Root Mean Squared Covariance" (RMSC):

$$\mathrm{RMSC} = \frac{\left|\left|\mathbf{ME^T}\right|\right|_F}{\sqrt{\mathrm{tr}(\mathbf{MM^T})\,\mathrm{tr}(\mathbf{EE^T})}} \tag{4}$$

Here, $||\quad||_F$ is the Frobenius norm and $\mathrm{tr}(\ )$ is the trace operator. In general, larger values of this metric imply a robust result while smaller values imply that the MCA modes do not capture a significant portion of the variance in the two datasets. For artificially generated data with similar dimensions to the data used here, poorly correlated fields yield RMSC ≈ 0.09 while well correlated fields yield RMSC ≈ 0.26, (this range will vary depending on the

size of the dataset and number of independent samples). The RMSCs computed for each of the study regions are: North Atlantic: 0.17, North Pacific: 0.17, South Atlantic: 0.16, South Pacific: 0.17, which are reassuringly high considering the large variability in the monthly MISR data. Finally, each of the MCA modes, which are shown in Fig. 3, indicate the percent covariance explained, and the percent variance explained in each of the two datasets by that mode. We discuss these modes in Section 5.

The combination of the MISR cloud fraction joint-histogram trends and associated spatial distribution ($\mathbf{U_M}$ and $\mathbf{V_M}$) can be projected back on to the original cloud fraction dataset to yield time series associated with each of the MCA modes. More precisely, this can be done by taking the outer product of a column of $\mathbf{V_M}$ and a (transposed) column of $\mathbf{U_M}$, which results in a matrix of the same size as $\mathbf{M}$. Each month of the original cloud fraction data can be restructured into a matrix of the same size as $\mathbf{M}$, and then the sum of the element-wise product of the outer product

of the column of $\mathbf{U_M}$ and $\mathbf{V_M}$ and each month of MISR cloud fraction data can be taken to retrieve a time-series. This time series is then standardized by subtracting its mean and dividing by its standard deviation. Each of the resulting time series based on the MCA modes will necessarily indicate trends because the MCA was performed on the trends computed using the original datasets.

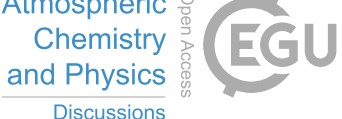

**Competing Interests:**

The authors declare that they have no conflict of interest.

**Acknowledgements:**

This research was supported by the MISR project at the NASA Jet Propulsion Laboratory (under contract 1318945),
and we wish to acknowledge the many contributions of the MISR team at JPL and the NASA Langley Research
Center Atmospheric Science Data Center without whom this project would not have been possible.



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



**Figures:**

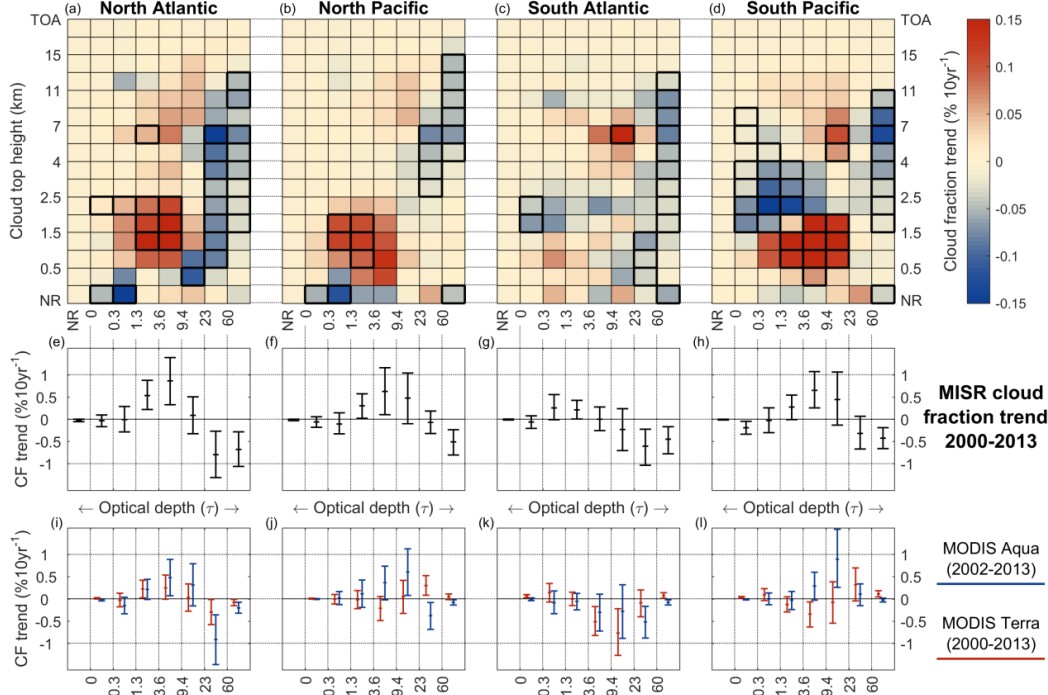

**Figure 1: Cloud fraction trends (2000-2013), in percentage per decade, computed for each bin of MISR cloud top height vs optical depth joint histograms (top, panels a-d). Trends are spatially averaged over the four extratropical ocean basins listed above each panel (and these regions are depicted in Figs. 2 and 3). Bold bordered bins denote trends that are significant at the 95% level. The middle set of panels (e-h) show spatially averaged trends in cloud fraction with respect to the optical depth bins only (summing over all cloud top height bins), with whiskers denoting 95% confidence intervals.**

**The bottom panels (i-l) show similar cloud optical depth trends for MODIS Aqua (blue) and Terra (red). The MODIS CTH-OD product uses slightly different optical depth bins and two of the MODIS high-optical-depth bins have been summed over to create one OD>60 bin for easier comparison to the MISR trends. MISR "No Retrieval" bins, denoted "NR," include cases where either the cloud top height or optical depth retrieval failed. MISR observed a significant reduction in cloud fraction in one or both of the $\tau > 23$ bins in each ocean basin, and three (all but the South Atlantic)**

**show a significant increase in low level cloud of moderate optical thickness. MODIS Aqua is largely consistent with MISR (showing increases in clouds of moderate optical thickness in the same three basins, and differs mostly in showing no statistically significant decrease in clouds with an optical thickness greater than 23 in the South Pacific). MODIS Terra is less consistent with MISR (and MODIS Aqua) in showing smaller decreases or slight increases in clouds with an optical thickness greater than 23, likely due to calibration issues (see Section 2 of text for discussion).**




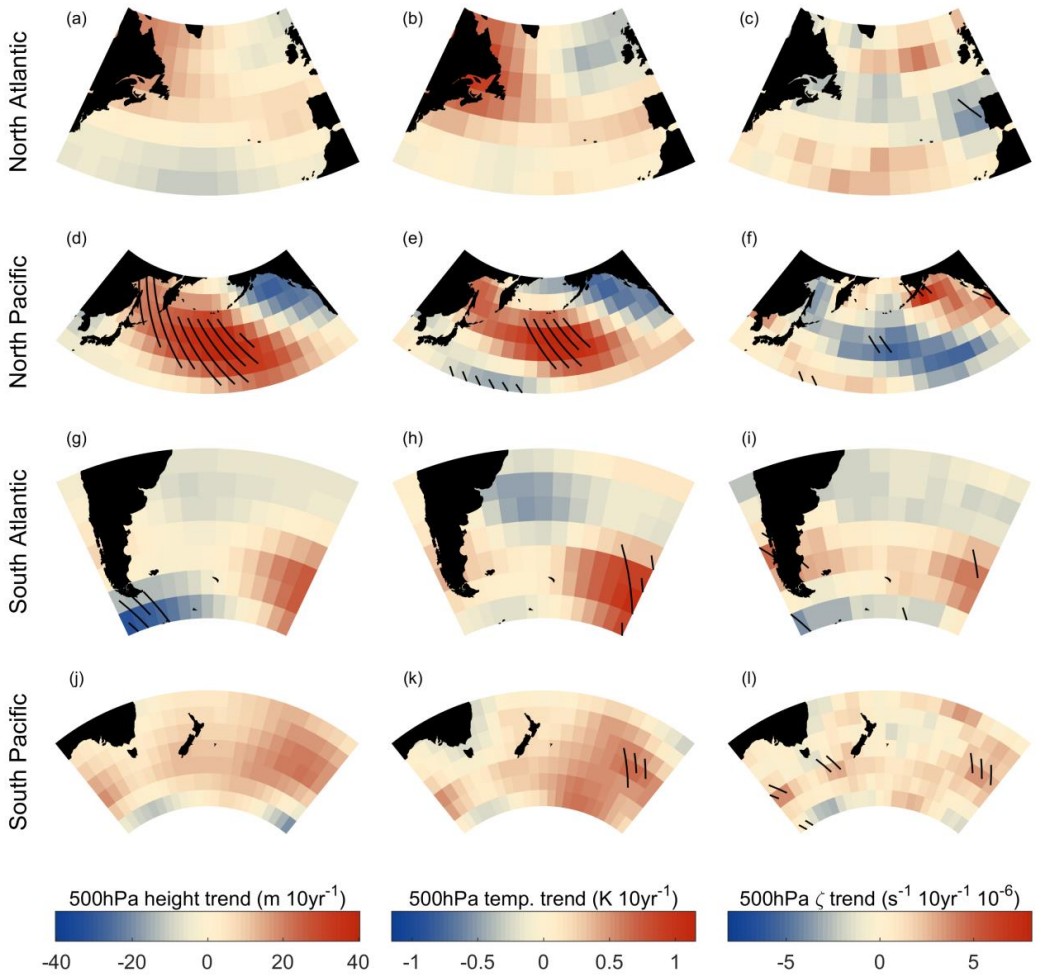

**Figure 2: Trends between 2000 and 2013 in ERA-Intrim 500hPa geopotential height, temperature, and absolute vorticity in each of the extratropical study regions: North Atlantic (25-65N, 280-360E), North Pacific (25-65N, 120-240E), South Atlantic (25-65S, 280-360E), and South Pacific (25-65S, 120-240E). Hashing denotes trends that pass a 95% confidence test. The North Pacific region shows significantly increased heights (d), temperature (e), and anticyclonicity (f) (negative trends in absolute vorticity in the northern hemisphere) primarily in the storm track regions and extending into the subtropics. All of these features are commonly associated with extratropical high pressure systems. Both of the southern hemisphere study regions show similar increases in heights (g and j), temperature (h and k) and anticyclonicity (i and j) in the storm track regions as well, however only small areas in the center of the locations with positive trends pass the significance test. The North Atlantic study region (a-c) shows no trends that are significant at the 95% level.**










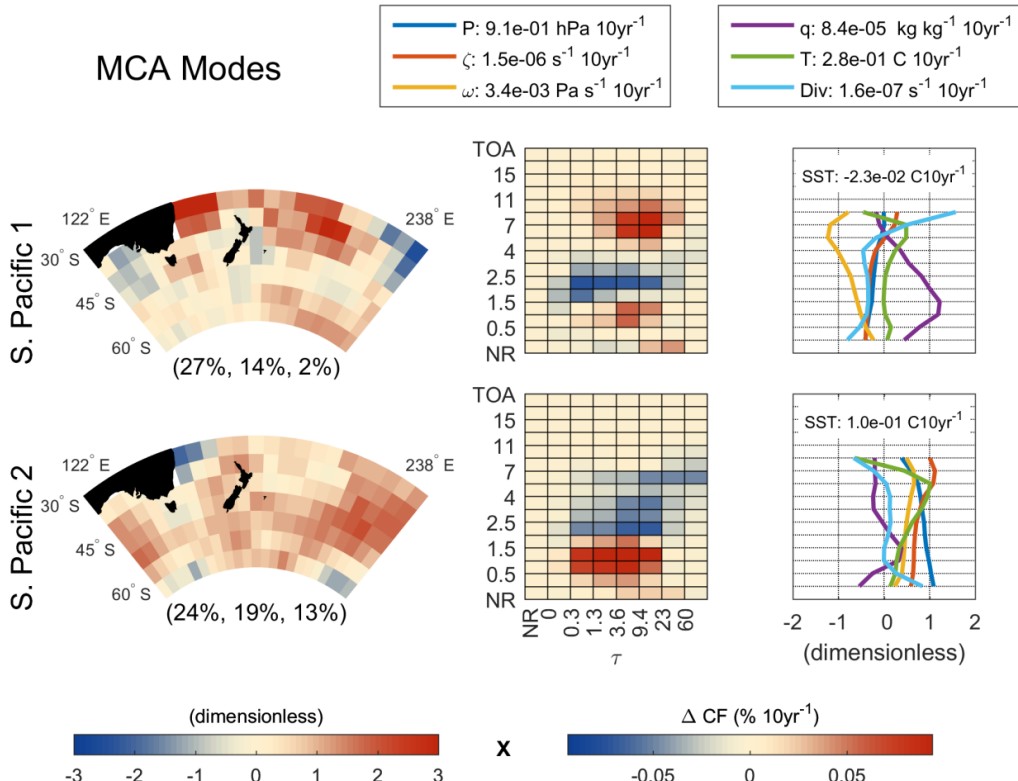

**Figure 3: Results of maximum covariance analysis between MISR cloud fraction and ECMWF reanalysis variable trends (2000-2013). The MCA modes are computed by applying a singular value decomposition to the spatial covariance matrix between MISR cloud fraction joint histogram trends and trends in vertical profiles of six ERA interim variables and sea surface temperature, as described in Appendix I. The middle column shows the pattern of changing cloud fraction in CTH-OD space ($U_M^*$ in Appendix I) and the right column shows the associated trends in the vertical profiles of the reanalysis variables ($U_E^*$). The left column shows the spatial loading patterns obtained by projection of the ECMWF modes on the far right on to the original trends. The spatial patterns are dimensionless, and the scales associated with the CTH-OD trend patterns are shown in the bottom right color-bar while the scales for the ERA-Interim profile trend patterns are shown in the legend near the top. The trend (due to each mode) at each latitude and longitude grid point is obtained by multiplication of the joint histogram or vertical profiles with the dimensionless spatial pattern on the left. Listed with each mode (in parentheses under the plot of the mode's spatial loading pattern) is the fraction of the covariance explained by that mode, followed by the fraction of variance in the ECMWF and MISR datasets, respectively. A 5-degree latitude/longitude grid was used.**







**Figure 4:** Correlation matrix between time series computed for each MCA mode shown in Fig. 3 and each of the CPC climate indices (a). The numerical values in the left panel (a) are correlation coefficients with the decimal point omitted, bold values indicate significance (p<0.01), coloring also denotes the sign and magnitude of the correlation. Only CPC indices with significant correlations with at least one MCA mode are included. The climate indices are sorted such that those that describe the most variance averaged across all MCA modes appear near the top. The right panel (b) shows normalized trends in the CPC indices between 2000-2013, along with their magnitude expressed in standard deviations, showing for example, that the Pacific Decadal Oscillation (PDO) index was trending lower between 2000-2013, while the North Pacific Index (NPI) and Antarctic Oscillation/Southern Annular Mode (AAO/SAM) index increased.







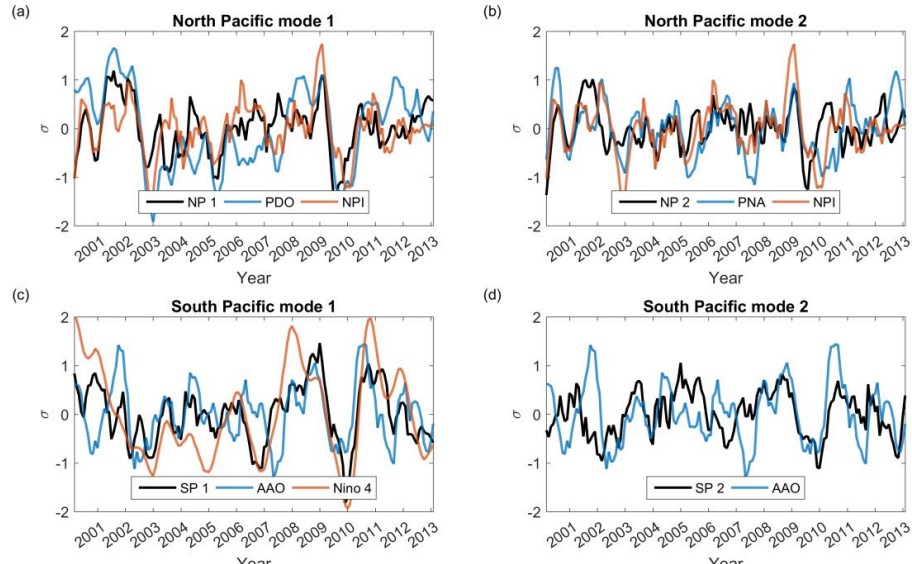

**Figure 5: Time series from the top two North and South Pacific MCA modes, shown along with the most correlated CPC indices identified in Fig. 3. The North Pacific Index is strongly connected to both North Pacific MCA modes (panels a and b), with correlations of .56 and .42 for modes 1 and 2 respectively. In the South Pacific, mode 1 (c) is strongly correlated with the Nino 4 index (-0.50) while mode 2 (d) is correlated with the Southern Annular Mode (0.32). In the plots above, the sign of each index that is negatively correlated with the MCA mode shown has been flipped and a 5-month boxcar filter has been applied to make the plots more readable. No filtering was applied to derive the correlation coefficients given (the correlations between all CPC indices and all MCA modes are given in Fig. 4 (a)).**






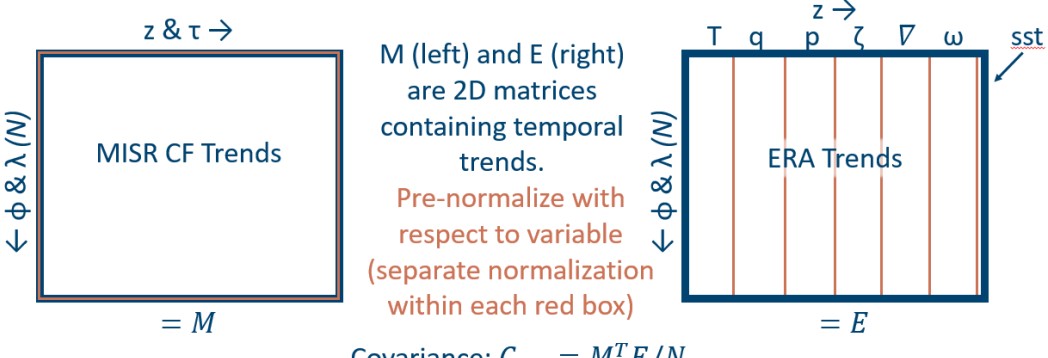

Figure 6: Graphical depiction of how the maximum covariance analysis was formulated. Each of the boxes represents a 2-dimensional data matrix. The rows of each matrix correspond to latitude and longitude coordinates. Matrix 'M' contains trends in the MISR cloud fraction joint histogram data. Matrix 'E' contains trends in the ECMWF reanalysis data. The columns of 'M' represent variability with respect to cloud optical depth and cloud top height. The columns of 'E' represent variability with respect to variable type and vertical level. Normalization (subtracting the mean and dividing by the standard deviation) is performed within each red box. $C_{ME}$ is then a covariance matrix that captures spatial covariance between the trends in the cloud fraction data and the reanalysis data.

