# Peer review of "Cloud responses to climate variability over the extratropical oceans as observed by MISR and MODIS"

_Atmospheric Chemistry and Physics, 2018_

## Referee Comment (RC1) · Anonymous Referee #1 · 5 Nov 2018

Summary: The authors perform a maximum covariance analysis (MCA) on joint trends in cloud fraction histograms and meteorological variables in four ocean basins. The analysis yields modes that contribute to the joint trends. Several of these modes are correlated with known modes of variability, such as ENSO and the PDO.

Comments: The analysis seems clean and the results are interesting, although I felt it was difficult at times to interpret the findings physically. Fig. 3 shows an almost overwhelming amount of information. My main comment would be to better link the modes in Fig. 3 to the trends in cloud fraction histograms in Fig. 1, see below.

- This isn't fundamental, but I was wondering why the analysis was restricted to pre-

[Figure]

2013?

- Justification for the number of modes in MCA analysis: you justify this later in the text (L382), but it would be better to explain from the start how many MCA modes you are retaining and why.

- I think a disadvantage of the analysis method is that each of the modes is assumed to have a fixed vertical structure. Only the sign and magnitude of the mode can vary spatially, but the structure of the anomaly pattern is set. I could imagine that, say, the PDO would have distinct vertical signatures in different parts of the North Pacific. Can the authors comment on this?

- It's hard to make the link between the cloud histogram trends introduced in Fig. 1 and the modes in Fig. 3. To what extent do the modes explain the total trends? In each basin, does the sum of the two or three modes retained in the analysis yield something similar to the total trend?

Other minor comments: L140: Can you motivate your choice of detrending the indices? If the trends in some of these indices explain part of the cloud trends that you're interested in, you might be losing part of the signal. For example, wasn't there a negative trend in ENSO during this period? L217,674: Hashing → hatching? L417: Change to "6. Conclusions"
* * *

---

## Referee Comment (RC2) · Anonymous Referee #2 · 15 Nov 2018

The authors present an analysis of cloud fraction trends as observed by MISR and MODIS satellite instruments over the period from 2000 to 2013. These trends are related to trends in meteorological quantities over the same period as represented in the ERA Interim reanalysis dataset. The analysis is primarily based on a Maximum Covariance Analysis, a method related to principal component or empirical orthogonal function analysis, but based on the covariance matrix between cloud fraction data and the meteorological quantities from ERA Interim.

The analysis is carried out in four ocean basins more or less independently - the North and South Atlantic, and the North and South Pacific. The trend analysis identifies an

increase in low cloud of moderate optical depth in three of the four basins (all but the South Atlantic), and a reduction in clouds of high optical depth in all basins, especially at high altitudes, although the authors raise the possibility that some of the latter trends could be a result of an unaccounted-for drift in the MISR calibration.

The connection of these trends to the meteorological quantities is, however, harder to interpret. Clear decadal changes in the North Pacific are identified over this period, but changes in the other ocean basins are less coherent. And while the covariance analysis identifies relationships between trends in cloud fraction and those in the meteorological fields that seem to be physically coherent, it's not clear to me what light this analysis shed on the trends.

So my main concern is methodological – not to say that it is wrong in any way; indeed it looks to have been quite compentently done, and it is well explained in the text. But the covariance analyasis seems to be creating more problems than it solves, in several senses. Firstly, the multivariate nature of the SVD means that the method is looking for vertical structures in all of the meteorological fields that are consistent across the whole basin. This isn't unreasonable, but it's not obvious to me that these structures should be spatially consistent: for instance the horizontal transport and shears will vary strongly across the basins. This means that you need a bunch of 'modes' in order to decompose a specific signal, which may go some way towards explaining modes like NP1 and NP2, or SA1 and SA2 that look fairly similar to each other in terms of their expression in the cloud fraction space and seem to be 'correcting' each other to some extent. Secondly, it's not obvious how to connect the identified 'modes' to the trends themselves – or how to connect them to the climate modes. Both of these issues might be avoided by using a simpler analysis.

This method may be better suited to looking at month-to-month variability, which will contain much more information than a single spatially varying map of decadal trends.

I think these challenges need to be better addressed for the work to be publishable.

Perhaps the method has advantages that I am not seeing - if so the authors should make this case. However, given that the conclusions drawn from this methodology are rather vague, I'm not sure this is the case.

To be more constructive, one reasonable question to address might be 'Can the cloud fraction trends be 'explained' by the trends in meteorological trends?' Or do we need to look to compositional changes? A more restricted version of this is to what extend can these trends be explained by decadal variability in the identified climate modes. If either of these could be answered clearly this would go some way towards answering the more fundamental question of why these trends are occurring, which the paper much more impactful.

I have a few additional specific comments but they are of relatively minor importance:

Trend significance – has a field significance test been considered? There are many trends being tested for significance here and individual significance tests for each trend may not be fully rigorous. (The trends look robust so I am reasonably convinced that they are not statistical artefacts, but one can be misled in these things.)

Has the seasonal dependence of the trends been considered?

Figures 4 and 5 are not very clearly discussed in the text – they are referred to some extent in the discussion of Figure 3, but there was clearly a lot of effort put into including a lot of information in these figures and the reader is left to extract this information on their own.
* * *

---

## Author Comment (AC1) · 26 Jan 2019

**Ref: acp-2018-520**

**Andrew Geiss and Roger Marchand**

**1-26-2019**

**Response to Reviewers**

Thank you to the editor and to the reviewers for the insightful comments, below we provide line-by-line responses to the reviewers' concerns. Reviewer comments are printed in *"quotations and italicized sans-serif."* Line numbers refer to the "tracked changes" version of the manuscript.

**Response to Reviewer 1's Comments:**

*"The analysis seems clean and the results are interesting, although I felt it was difficult at times to interpret the findings physically. Fig. 3 shows an almost overwhelming amount of information. My main comment would be to better link the modes in Fig. 3 to the trends in cloud fraction histograms in Fig. 1, see below."*

We have provided some additional discussion of Fig. 3 in Section 5. See below (line 57 this document) concerning linking Fig. 3 and Fig. 1.

*"This isn't fundamental, but I was wondering why the analysis was restricted to pre-2013?"*
This work began in 2014 and that is when the MISR data was retrieved. Our ongoing work comparing the MISR CTH-OD dataset to monthly and inter-annual climate variability uses a
longer dataset.

*"Justification for the number of modes in MCA analysis: you justify this later in the text (L382), but it would be better to explain from the start how many MCA modes you are retaining and why."*
Discussion has been added to Section 4, line 299.

*"I think a disadvantage of the analysis method is that each of the modes is assumed to have a fixed vertical structure. Only the sign and magnitude of the mode can vary spatially, but the structure of the anomaly pattern is set. I could imagine that, say, the PDO would have distinct vertical signatures in different parts of the North Pacific. Can the authors comment on this?"*

This is a good point. The vertical profiles in each individual MCA mode are fixed and this is inherent to our MCA formulation. We note that most regions (the North Pacific included) are impacted by different modes with each mode having a different loading pattern. In a holistic sense the blending (or relative weight) of each mode carries information about the vertical variations. Nonetheless, we agree with the
point, that the technique does NOT capture horizontal (spatial) variations in the vertical structure within a single mode. We agree this deserves treatment in the text and have added discussion at line 297.

It is perhaps worth noting that other similar linear decomposition techniques such as EOF or correlation analysis are typically applied only at single pre-selected vertical levels. We could (of course) apply the
MCA to individual vertical levels, but this would create a morass of modes (a unique set of modes for each level) and this would likely require application of additional correlative or dimensional-reduction techniques to interpret.

We would like to stress here that our goal in this project (as stated in the introduction) was to "explore linkages" between the MISR extratropical cloud fraction trends and ERA-Interim reanalysis variables, and we think the MCA approach has allowed us to do this. In particular, we have learned from this exercise that in furthering our understanding of observed changes in mid-latitude clouds we should focus on changes related to high pressure (anticyclonic) systems, the PDO, SAM, and ENSO in future studies. In fact, we are in the process now of doing additional regression analysis that more directly addresses the impact of the PDO, SAM and ENSO, on clouds and this analysis will allow for an examination of variations in the vertical structure.

*"It's hard to make the link between the cloud histogram trends introduced in Fig. 1 and the modes in Fig. 3. To what extent do the modes explain the total trends? In each basin, does the sum of the two or three modes retained in the analysis yield something similar to the total trend?"*

Agreed, another good point. We have added Table 1 on page 25, and mathematical description on line 565 in Appendix I, which estimates the amount of the total cloud, low cloud, and thick cloud trend explained by each MCA mode in the spatial-average for each ocean basin (as shown in Figure 1). These are computed by taking the product of the cloud trend pattern and associated loading pattern for each MCA mode, computing the basin average and subtracting from the trends shown in Figure 1. The fractional trend explained for each mode is computed as:

$$\frac{\left(\sum_{jhist} |\Delta_{MCA} - \Delta_{obs}|\right)}{\sum_{jhist} |\Delta_{obs}|}$$

where the summations are over the bins in the cloud occurrence histograms and the deltas are the spatially averaged trend in cloud fraction in that bin. An important thing to note about this estimate is that in the MCA, SVD is applied to the spatial covariance matrix and the loading patterns are computed by a projection of the resulting MCA patterns on to the cloud fraction and reanalysis trends. This means that unlike classical EOF analysis, there is no orthogonality constraint on the spatial loading patterns, and consecutive MCA modes can theoretically explain overlapping parts of the same trend, so the percentages are not necessarily cumulative across multiple modes. Instead each percentage should be interpreted individually as: "how much of the observed trend can a single MCA mode explain by itself?" Here, the cloud (joint-histogram) loading pattern is projected on to the spatial loading pattern derived from the reanalysis data, so this is also an estimate of the strength of the meteorology-cloud relation identified by the mode.

*"L140: Can you motivate your choice of detrending the indices? If the trends in some of these indices explain part of the cloud trends that you're interested in, you might be losing part of the signal. For example, wasn't there a negative trend in ENSO during this period?"*

We would like to identify potentially causal relationships between the CPC indices and the MCA modes, and examine this by computing the correlation coefficients shown in Figure 4. If the signals are not detrended then many of the index-mode pairs will have non-zero correlations just by virtue of both signals containing a trend. For instance, the PDO might show correlation with the South Atlantic modes just because both signals happened to either increase or decrease during the period studied even if there is no physical connection. Instead we compute these correlations after first detrending, and then show the strength of the trend that was removed in the right panel in Fig. 4.  In effect, our correlation coefficients show the strength of the relationship based on co-variability at monthly time scales.  We have added text to make this clear: line 140 & 332.

*"L217,674: Hashing→hatching? L417: Change to: "6. Conclusions""*

Changed, now lines 219 and 754.

**Response to Reviewer 2's Comments:**

It seems that the primary concerns expressed here are with the decision to apply MCA and interpretation of the MCA results: The primary motivation for applying MCA is the desire to comprehensively compare many cloud type variables (each bin the CTH-OD histogram) and many meteorological variables (6 variables at 16 heights). While we could choose individual cloud type / reanalysis variable pairs and compare them one at a time using regression analysis or similar techniques, it is not clear a-priori which pairs are the most important to examine.  And likewise, examining all possible combinations would produce a massive set of relationships and require some additional dimensionality-reduction techniques to interpret. The MCA allows us to compute a covariance matrix (a comparison of every cloud/meteorological variable pair) and then identify the strongest sources of covariance between the two datasets.

We would like to stress here that our goal in this project (as stated in the introduction) was to "explore linkages" between the MISR extratropical cloud fraction trends and ERA-Interim reanalysis variables, and we think the MCA approach was a good choice for such an exploration. In particular, we have learned from this exercise that in furthering our understanding of observed changes in mid-latitude clouds we should focus on changes related to high pressure (anticyclonic) systems, the PDO, SAM, and ENSO in future studies.

Regarding comparison of cloud trends to CPC indices: Direct comparison of the cloud to the climate indices using temporal covariances (or regression maps) is a sensible approach to the degree one is seeking to understand how any specific CPC index relates to cloud cover.  In fact, we are in the process now of doing additional regression analysis that more directly addresses the impact of the PDO, SAM and ENSO, and we plan to publish these results in a later manuscript.

We provide some additional justification for the use of MCA starting on line 260 in the manuscript.

*"The authors present an analysis of cloud fraction trends as observed by MISR and MODIS satellite*

*instruments over the period from 2000 to 2013. These trends are related to trends in meteorological quantities over the same period as represented in the ERA Interim reanalysis dataset. The analysis is primarily based on a Maximum Covariance Analysis, a method related to principal component or empirical orthogonal function analysis, but based on the covariance matrix between cloud fraction data and the meteorological quantities from ERA Interim. The analysis is carried out in four ocean basins more or less*

*independently - the North and South Atlantic, and the North and South Pacific. The trend analysis identifies an increase in low cloud of moderate optical depth in three of the four basins (all but the South Atlantic), and a reduction in clouds of high optical depth in all basins, especially at high altitudes, although*

*the authors raise the possibility that some of the latter trends could be a result of an unaccounted-for drift in the MISR calibration."*

*"The connection of these trends to the meteorological quantities is, however, harder to interpret. Clear decadal changes in the North Pacific are identified over this period, but changes in the other ocean basins are less coherent."*

This comment seems to be in reference to Figure 2. Panels d-f in Figure 2 show strong trends in 3 meteorological fields in the North Pacific that pass a 95% confidence test at many locations in the region. The other 3 ocean basins show weaker trends in the meteorological fields.  It is true that the changes outside of the North Pacific, do not pass a 95% confidence test for a statistically significant trend.  But that does not mean there was no change in the meteorological fields. To
the degree the ERA data are correct, there are real and coherent patterns of change that are consistent across multiple meteorological fields.  Note that the panels in this figure share a color scale and the trends shown in other ocean basins are only marginally smaller than those observed in the North Pacific. The weakest trends occur in the North Atlantic, where changes in the storm track region are perhaps 50% of the magnitude of the trends in the North Pacific. That the
changes in the other basins do not pass the trend test ONLY means that we can not be confident that the changes represent a true trend given the natural variability.

        The MCA analysis demonstrates that the changes in the ERA meteorology (whether or not they are trends) are correlated with changes in the MISR clouds.  Indeed, the cloud changes in the
North Atlantic (MCA mode NA1) are remarkably similar to those in the North Pacific (MCA mode NP2). Our interpretation is that the meteorological trends in the North Atlantic, and other regions during this time period are weak when compared to the natural monthly variability in the extratropics, but likely represent a physical change, and are strong enough to cause a measurable change in cloudiness in the region. This is discussed in the text, line 240.
        *"And while the covariance analysis identifies relationships between trends in cloud fraction and those in the meteorological fields that seem to be physically coherent, it's not clear to me what light this analysis shed on the trends."*

In our revisions we have added Table 1 which estimates how much of the basin averaged trend (Figure 1) can be explained by individual MCA modes (see response to reviewer #1, above on line 57 this document). Furthermore, the MCA identifies specific meteorological changes that tend to be co-located with specific cloud changes in each basin, and we think that the results are useful beyond their ability to explain the basin-averaged trends in Figure 1. For instance, the
MCA results can be used to make statements such as: "a shift towards high pressure regimes in the northern hemisphere storm track regions is related to increased low-altitude cloud occurrence and reduced high-altitude optically thick cloud (NA1, NP2)," and this alone is an insight into the interaction of cloud and meteorology on multi-year time scales.

*"So my main concern is methodological – not to say that it is wrong in any way; indeed it looks to have been quite compentently done, and it is well explained in the text. But the covariance analyasis seems to be creating more problems than it solves, in several senses. Firstly, the multivariate nature of the SVD means that the method is looking for vertical structures in all of the meteorological fields that are consistent across the whole basin. This isn't unreasonable, but it's not obvious to me that these structures*

*should be spatially consistent: for instance the horizontal transport and shears will vary strongly across the basins. This means that you need a bunch of 'modes' in order to decompose a specific signal, which may go some way towards explaining modes like NP1 and NP2, or SA1 and SA2 that look fairly similar to each other in terms of their expression in the cloud fraction space and seem to be 'correcting' each other to some extent."*

We agree that a significant limitation of the MCA is that variability in the vertical structure is not captured within the context of a single mode. Reviewer 1 makes the same point. Please see our response to reviewer#1 above, on line 31 in this document.

*Secondly, it's not obvious how to connect the identified 'modes' to the trends themselves*

Agreed. We have added Table 1 to the paper, which provides estimates of what fraction of the trends shown in Figure can be explained by the MCA modes. See the commentary above on line 57 in this document.

*– or how to connect them to the climate modes. Both of these issues might be avoided by using a simpler analysis. This method may be better suited to looking at month-to-month variability, which will contain much more information than a single spatially varying map of decadal trends. I think these challenges need to be better addressed for the work to be publishable. Perhaps the method as advantages that I am*

*not seeing - if so the authors should make this case. However, given that the conclusions drawn from this methodology are rather vague, I'm not sure this is the case."*

Figures 4 and 5 use correlation coefficients to relate the MCA modes to the known modes of climatic variability and we do not see an issue with this per se.

In regards using a "simpler analysis", the answer (as per our comments at the start of this response) is absolutely yes. For example, one of our major conclusions is that much of the change in MISR clouds is a result of increased geopotential heights and anticyclonicity in all four oceanic basins. Presumably a simpler analysis looking at covariance between 500 hPA

geopotential height and MISR low cloud amount will reveal this same broad result. Likewise, we could have assumed the PDO was a major driver and sought to isolate the contribution of the PDO via a regression analysis at the start. But our intent was to explore the relationship between the MISR cloud changes and many ERA variables. The advantage of the MCA is not that it is simpler to implement or easier to interpret. Rather the advantage of the technique is that we did not need to choose a specific meteorological field or a particular subset of the MISR CTH-OD clouds. Rather the MCA allows us to examine the co-variance of many meteorological variables simultaneously, and to identify coherent relationships between all of these variables.

*"To be more constructive, one reasonable question to address might be 'Can the cloud fraction trends be 'explained' by the trends in meteorological trends?' Or do we need to look to compositional changes? A more restricted version of this is to what extend can these trends be explained by decadal variability in the identified climate modes. If either of these could be answered clearly this would go some way towards answering the more fundamental question of why these trends are occurring, which the paper much more*

*impactful."*

Again, see our newly added Table 1 on page 25. This provides an estimate of how much of the cloud trends can be explained by the MCA modes derived from the reanalysis data.

The MCA makes clear is that there is no single meteorological field or mechanism that explains all of the cloud changes in each region. Furthermore, different cloud-meteorology interactions have distinct spatial structure. The cloud trends shown in Figure 1 are a combined effect of all of these spatial structures and mechanisms, meaning that the MCA has shown something that cannot easily be shown by performing 1-to-1 comparisons between a cloud category and a meteorological field or climate index. Part of our ongoing work involves directly assessing the relationship between some of these climate indices and cloud occurrence by examining monthly covariances, and we plan to publish these results in a later manuscript.

*"I have a few additional specific comments but they are of relatively minor importance: Trend significance*

*– has a field significance test been considered? There are many trends being tested for significance here and individual significance tests for each trend may not be fully rigorous. (The trends look robust so I am reasonably convinced that they are not statistical artefacts, but one can be misled in these things.)"*

We considered but did not apply field significance tests. The fields analyzed are small enough that typical field significance tests that account for spatial autocorrelation (e.g. Bretherton et al. 2000, "The effective number of spatial degrees of freedom of a time varying field") are not well suited, and as you mention, most of the trends appear to be very spatially coherent, so we are not particularly concerned that the trends are non-physical.

*"Has the seasonal dependence of the trends been considered?"*

Yes, we examined the seasonal dependence of the trends early on in this project, but the differences between seasons were relatively small and we chose not to include these results in the paper. We've added a comment to this effect in the paper on line 166.

*"Figures 4 and 5 are not very clearly discussed in the text – they are referred to some extent in the discussion of Figure 3, but there was clearly a lot of effort put into including a lot of information in these figures and the reader is left to extract this information on their own."*

Agreed. We have added some additional discussion of these figures to Section 4 starting on line 332.